# Scale-Resolving Simulation of a Propane-Fuelled Industrial Gas Turbine Combustor Using Finite-Rate Tabulated Chemistry

**Kai Zhang, Ali Ghobadian and Jamshid M. Nouri \***

Department of Mechanical Engineering and Aeronautics, University of London, Northampton Square, London EC1V 0HB, UK; Kai.Zhang.1@city.ac.uk (K.Z.); A.Ghobadian@city.ac.uk (A.G.)

\* Correspondence: j.m.nouri@city.ac.uk; Tel.: +44-(0)20-7040-8119

**Abstract:** The scale-resolving simulation of a practical gas turbine combustor is performed using a partially premixed finite-rate chemistry combustion model. The combustion model assumes finite-rate chemistry by limiting the chemical reaction rate with flame speed. A comparison of the numerical results with the experimental temperature and species mole fraction clearly showed the superiority of the shear stress transport, K-omega, scale adaptive turbulence model (SSTKWSAS). The model outperforms large eddy simulation (LES) in the primary region of the combustor, probably for two reasons. First, the lower amount of mesh employed in the simulation for the industrial-size combustor does not fit the LES's explicit mesh size dependency requirement, while it is sufficient for the SSTKWSAS simulation. Second, coupling the finite-rate chemistry method with the SSTKWSAS model provides a more reasonable rate of chemical reaction than that predicted by the fast chemistry method used in LES simulation. Other than comparing with the LES data available in the literature, the SSTKWSAS-predicted result is also compared comprehensively with that obtained from the model based on the unsteady Reynolds-averaged Navier–Stokes (URANS) simulation approach. The superiority of the SSTKWSAS model in resolving large eddies is highlighted. Overall, the present study emphasizes the effectiveness and efficiency of coupling a partially premixed combustion model with a scale-resolving simulation method in predicting a swirl-stabilized, multi-jets turbulent flame in a practical, complex gas turbine combustor configuration.

**Keywords:** CFD; combustion modeling; gas turbine combustor; scale-resolving simulation; partially premixed flame

---

## 1. Introduction

The need for effective and efficient mathematical descriptions of turbulent reactive flows has led to significant research interest in turbulent models that rely on resolving the turbulent spectrum [1–3] and combustion models [4,5], which account for comprehensive chemical reactions without sacrificing too much computational power. The LES approach [6] is known as one of the most popular spectrum-resolving-based turbulent models, thanks to one of its most significant features. That is to say, the large-scale turbulent eddies which dominate the turbulent dynamics and motions are directly resolved by invoking a cell-size-based filtering technique [7]. However, this well-known superiority of LES in describing large eddy motions also leads to one defect of the model: the high computational power required for LES simulation due to the employment of small mesh cells to fill the computational domain. Therefore, the LES modeling approach is more commonly used in academia rather than industry, because of the complex and large-scale components often required in the latter. In academia, on the other hand, the primary focus is the physical phenomena in small to medium scale configurations, allowing the use of a more computationally expensive LES approach.

Because of the LES defect, alternative mathematical methods, such as the detached eddy simulation (DES) [8] and the scale adaptive simulation (SAS) [9], have been attracting more attention from industries due to their lower requirement of computational powers and their ability to resolve the turbulent spectrum. More specifically, the DES method employs a Reynolds-averaging Navier–Stokes (RANS)-based modeling approach near the boundary layer regions, and the LES modeling approach in the mainstream regions [10]; furthermore, the SAS method invokes an extra von Karmen length scale in order to ensure an accurate energy distribution along the turbulent spectrum, and to improve the performance of the RANS model [11]. In fact, the DES is often considered a first generation hybrid RANS-LES modeling method, and the SAS is considered the second generation. This is because, compared to DES, the SAS has two extra advantages allowing it to be quickly adapted to industrial needs. First, the SAS has no explicit local grid spacing dependency in each direction, unlike the DES [12]. Second, the SAS can be easily incorporated into the existing experimentally adapted RANS model.

Consequently, Mentor and Egorov [13–18] studied the accuracy of the SAS method under different flow conditions. For example, the 3D acoustic cavity and ITS (Institut für Thermische Strömungsmaschinen, Stuttgart) turbulent combustion were investigated using the SSTKWSAS model (the incorporation of the SAS approach into the shear stress transport K-omega model) [15]. Good agreements between numerical and experimental results were obtained under the condition of using coarse mesh. It was also shown [18] that the SAS outperforms LES in a periodic hill flow simulation (the one described by Frohlich et al. [19]). Due to the coarse mesh and large time step used, the LES returned incorrect results, while the SAS provided trustworthy results that were less dependent on mesh or time step conditions. Many other case studies were provided by Egorov et al. [20], demonstrating the good performance of SAS for various flows.

However, despite the accuracy of the SAS method in mathematically describing turbulent flow motions, it is less valid for very complex reactive flows. One typical study might be the recent work of Lourier et al. [21]. The thermoacoustic instability in a partially premixed lean swirl combustor was investigated with the SSTKWSAS model coupled with two combustion models: the eddy dissipation concept (EDC) model [22] and a finite-rate detail chemistry model incorporating an assumed probability density function (PDF) method for subgrid modeling [23]. The numerically obtained velocity, temperature and mixture fraction field fitted reasonably well with the experimental data of Meier et al. [24]. The same turbulent model was also applied to the simulation of the ITS combustion chamber fitted with a swirler [20]. The improved performance of SSTKWSAS over the traditional shear stress transport, k-omega (SSTKW) model was observed on an unstructured mesh.

In the present study, a propane-fuelled, partially premixed industrial combustor is numerically simulated with the SSTKWSAS model to further demonstrate the effectiveness and efficiency of the model. The work differs from previous studies and provides extra contributions to the mathematical modeling community in terms of the following points:

(1) The investigated combustor includes important features of a realistic industrial-used gas turbine combustor, such as the multiple fuel holes, the swirler, the primary and secondary holes, the porous walls, etc.;

(2) The SSTKWSAS approach is coupled with a finite-rate tabulated chemistry-based combustion model validated in our previous work [25];

(3) Comprehensive comparisons of the SSTKWSAS model-produced results with experimental data are provided for the main flame region of the combustor;

(4) The performances of different turbulence models are discussed and compared with LES results in the literature [26].

Overall, this study is motivated by the lack of detailed SSTKWSAS (coupled with a valid combustion model) simulation for a practical industrial-used gas turbine combustor. The extended flame residence time in a realistic gas turbine combustor differs greatly from those in simple burners or combustors, thus requiring the application of more comprehensive mathematical models.

## 2. Mathematical Models

### 2.1. Turbulent Modeling

In this study, three turbulent models are employed to simulate the chosen gas turbine combustor. The shear stress transport K-omega (SSTKW) [27] and Reynolds stress transport (RSM) [28] models are RANS-based models widely used for industrial applications. The SSTKWSAS model is a scale-resolving model derived from the SSTKW model by invoking a SAS approach [15]. The Reynolds-averaged transport equations for SSTKW and SSTKWSAS are closed by solving two transport equations for turbulent kinetic energy, $k$, and vortex frequency, $\omega$. The RSM model is closed by solving six transport equations for Reynolds stresses.

Although SSTKWSAS and SSTKW share almost the same equations, as shown in Equations (1) and (2), their performances differ mainly because of the involvement of an extra term, $Q_{SAS}$, in $\omega$ equations, as expressed in Equation (3).

$$\frac{\partial \rho k}{\partial t} + \frac{\partial \rho k u_i}{\partial x_i} = \frac{\partial}{\partial x_j}\left[\mu_{eff}\frac{\partial k}{\partial x_j}\right] + G_k - D_k \tag{1}$$

$$\frac{\partial \rho \omega}{\partial t} + \frac{\partial \rho \omega u_j}{\partial x_i} = \frac{\partial}{\partial x_j}\left[\mu_{eff}^*\frac{\partial \omega}{\partial x_j}\right] + G_\omega - D_\omega + E_\omega + Q_{SAS} \tag{2}$$

$$Q_{SAS} = max[\rho C_1 \kappa S^2 (\frac{L_t}{L_{vk}})^2 - \frac{2\rho k}{C_2}max(\frac{1}{\omega^2}\frac{\partial \omega}{\partial x_j}\frac{\partial \omega}{\partial x_j}, \frac{1}{k^2}\frac{\partial k}{\partial x_j}\frac{\partial k}{\partial x_j}), 0] \tag{3}$$

where $G_k$ and $G_\omega$ represent the generation of $k$ and $\omega$ respectively. $D_k$ and $D_\omega$ represent the dissipation of $k$ and $\omega$. $E_\omega$ contains the cross-diffusion term due to transformation from the standard K-epsilon to K-omega model. $C_1$ and $C_2$ are model constants [29], and the turbulent length scale $L_t = \sqrt{k}/C_\mu^{1/4} \times \omega$. The von Karman length scale, $L_{vk}$, is proportional to the ratio between the first and second velocity derivatives. To provide high wave damping [18], the von Karman length scale is constrained by the mesh size $\Delta = \sqrt[3]{CV}$, where $CV$ is the control volume. $\rho$, $\mu_{eff}$, $\kappa$ and $S$ are the fluid density, effective viscosity, von Karman constant of 0.41 and scalar invariant of strain rate tensor, respectively.

For coarse mesh simulation using SSTKWSAS, the URANS (unsteady RANS) solution is obtained because of the high local von Karman length scale, i.e., the small $Q_{SAS}$ term leads to accumulated viscosity. For fine mesh simulation, the term allows the resolving of the turbulent spectrum up to the high wavenumbers with respect to the smallest grid spacing, and the model behaves like LES. For the present study, a coarse mesh and large time step have been used in order to check the accuracy of the SSTKWSAS model operating under URANS conditions. This greatly reduces the computational time required for an industrial, complex gas turbine simulation, compared to the time needed for LES simulation in the literature [26].

### 2.2. Combustion Modeling

The combustion simulation relies on the partially premixed finite-rate chemistry combustion model, ZTFSC (Zimont Turbulent Flame Speed Closure) [25], which has been proven to be more accurate than the non-premixed infinitely fast chemistry combustion models [26]. The superior performance of ZTFSC over other mathematical models is attributed to the control of the flame and flow dynamics by the important gas turbine features, such as the swirler and primary holes. The flame residence time in the main flame region of a practical gas turbine combustor is much greater than that of other simplified combustors, leading to a possible co-existence of burnt and unburnt mixture packages. Therefore, a flame front tracking method is needed to detect the local mixture status.

The mixture fraction theory [4] is employed to transform the flame characteristics from a real physical space to an imaginary mixture fraction space, for the purpose of using comprehensive chemical reactions [30] simplified with the steady laminar flamelet modeling (SLFM) method [31]. The ZTFSC

combustion model was originally designed for premixed flame modeling [32], and is blended with non-premixed models to form a partially premixed combustion model. Following the mixture fraction theory and ZTFCS, two transport Equation (4) are numerically solved with Equation (2) added to remedy the defect of Equation (1). This is because the first equation has no explicit introduction of the chemical reaction term $\dot{\omega}_c$, such that the chemical reactions are essentially infinitely fast. To limit the chemical reaction rate, the chemical reaction term is introduced in Equation (2). Equation (5) is then used to calculate interested reacting flow quantities ($q$) such as temperature, species fractions, etc.

$$\begin{cases} \frac{\partial \rho \theta}{\partial t} + \nabla \cdot (\rho u_k \theta) = \nabla \cdot (\rho \alpha_\theta \nabla \theta) \\ \frac{\partial \rho C}{\partial t} + \nabla \cdot (\rho u_k C) = \nabla \cdot (\rho \alpha_c \nabla C) + \rho \dot{\omega}_c \end{cases} \tag{4}$$

$$q = C \int_0^1 \varnothing_b(\theta) p(\theta) d\theta + (1 - C) \int_0^1 \varnothing_u(\theta) p(\theta) d\theta \tag{5}$$

where $p(\theta)$ represents a presumed beta PDF to account for the turbulent chemistry interaction [33] given in Equation (6). The constants $a$ and $b$ are calculated by introducing an extra transport equation for the variance of the mixture fraction.

$$P(\theta) = \frac{\Gamma(a+b)\theta^{a-1}(1-\theta)^{b-1}}{\Gamma(a)\Gamma(b)} \tag{6}$$

In Equation (5), when the progress variable (or reaction progress) $C$ is equal to 1, the mixtures are fully burnt and are solved by invoking an SLFM method. When $C$ equals 0, the mixed but unburnt mixtures are solved by simply using non-reacting mixture fraction theory. The chemical reaction rate, $\dot{\omega}_c$, is closed and modeled following the work in [32], given as

$$\rho \omega_c = \rho_u U_t |\nabla C| \tag{7}$$

## 3. Experimental Conditions and Numerical Methods

The simulated Tay-model combustor is identical to that described by Bicen et al. [34], representing a practical industrial-used can type gas turbine combustor (see Figure 1). The combustor includes the important components of a practical gas turbine combustor, such as 10 fuel holes, a combustor liner (porous wall), a swirler, 6 primary holes and 6 secondary or dilution holes. The diameters of the primary and secondary holes were originally 10 mm and 20 mm, but the diameter for the former is reasonably scaled to 8.6 mm in the present study, consistent with that introduced by Zhang et al. [25]. This strategy was introduced to compensate for the plug flow assumption for the six primary holes [35].

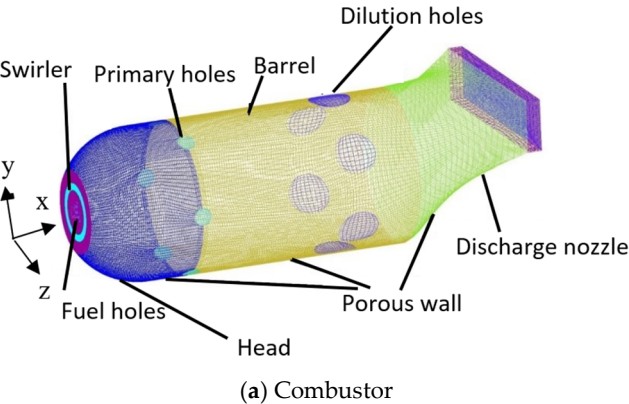

(**a**) Combustor

**Figure 1.** *Cont.*

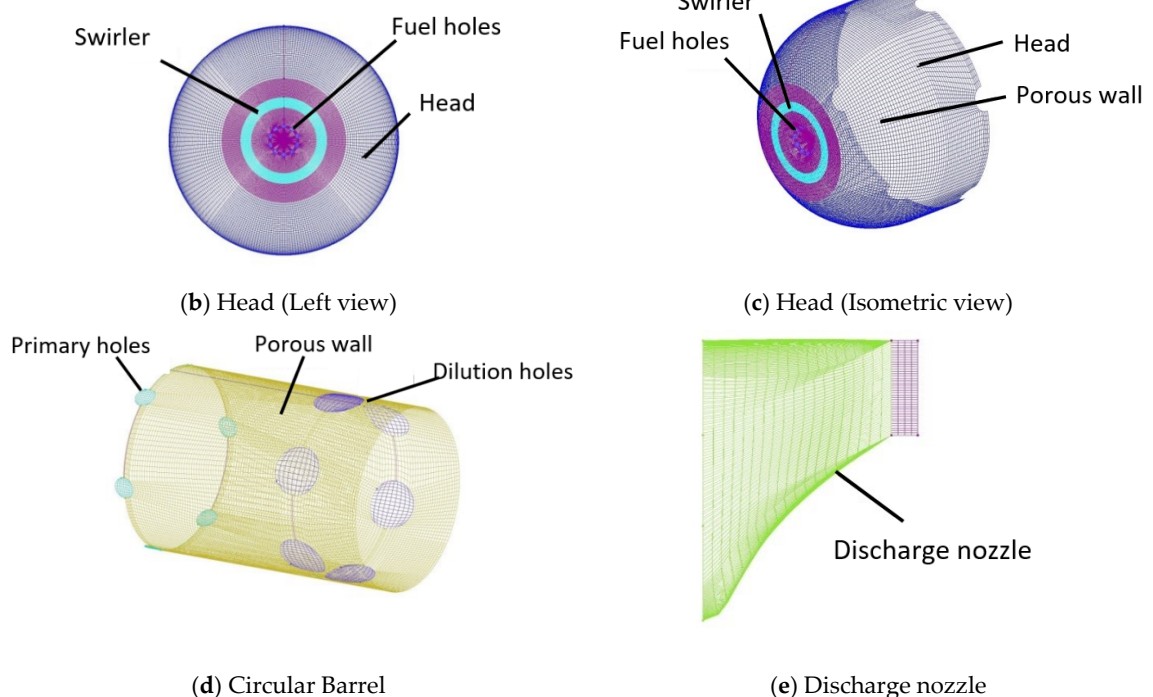

(**b**) Head (Left view)

(**c**) Head (Isometric view)

(**d**) Circular Barrel

(**e**) Discharge nozzle

**Figure 1.** Computational mesh.

The swirler vanes are simulated via computing axial and tangential velocity components, based on the effective area of the annular shape in Figure 1b. Following the suggestion of Crocker et al. and Lefebvre [36,37], Equation (8) is used to calculate tangential velocity.

$$\omega = \eta \frac{W_{sw}}{A_{se}\rho C_d (1-b)} tan\alpha \tag{8}$$

where the blockage factor $b$, turning efficiency $\eta$ and discharge coefficient $C_d$ are 0.1, 0.92 and 0.75, respectively. More details about the swirler can be found in the work of Bicen and Palma [38]. The swirl number of 1.01 for the straight vane swirler is obtained using Equation (9), corresponding to a high swirling strength.

$$S = \frac{2\left(1 - \left(\frac{d1}{d2}\right)^3\right)}{3\left(1 - \left(\frac{d1}{d2}\right)^2\right)} tan\theta' \tag{9}$$

The swirler inner and outer diameter $d_1$ and $d_2$ are 10.64 mm and 28 mm, respectively. The $\theta'$ is the geometric real angle without the influence of blockage factors, the discharge coefficients and the turning efficiencies. Hence, the swirler angle chosen for representing the swirler and its associated swirl effect is 54°, corresponding to the ratio between the flow tangential component (0.809) and the axial component (0.588).

A summary of the experimental conditions used for the present study is provided in Table 1, and the numerical boundary conditions are summarized in Table 2. The flow condition leads to a Reynolds number of 79,000, based on the mass flow rate of air and the diameter of the barrel.

**Table 1.** Experimental condition.

| Experiment | $\dot{M}_{air}$ (Kgs) | $\dot{M}_{propane}$ (Kgs) | Swirler Vane Angle | $P$ (atm) | $T_{inlet}$ (K) | Air-to-Fuel Ratio AFR |
|:---:|:---:|:---:|:---:|:---:|:---:|:---:|
| 1 | 0.1 | 0.00176 | 45° | 1 | 315 | 57 |

**Table 2.** Numerical boundary conditions.

| | Primary Jets | Dilution Jets | Swirler Jets | Fuel Jets | Porous Wall Jet (Head) | Porous Wall Jet (Barrel) | Porous Wall Jet (Nozzle) |
|---|---|---|---|---|---|---|---|
| $\dot{M}$ (Kg/s) | 0.0136 | 0.0533 | 0.0069 | 0.00176 | 0.0066 | 0.0138 | 0.0058 |
| T (K) | 315 | 315 | 315 | 315 | 315 | 315 | 315 |
| $n_{O2}/n_{N2}$ | 0.21/ 0.79 | 0.21/ 0.79 | 0.21/ 0.79 | - | 0.21/ 0.79 | 0.21/ 0.79 | 0.21/ 0.79 |
| $n_{C3H8}$ | - | - | - | 1 | - | - | - |

In terms of the numerical schemes, bounded second order implicit temporal and central differencing (BCD) [39,40] momentum discretization schemes are used for the SSTKWSAS modeling, because the model is able to push eddy viscosity down to the limit of grid resolution. This requires a non-dissipative BCD scheme to ensure all dissipations are produced from the turbulent model itself. The second order upwind [41] scheme is used for the discretizing reaction progress or progress variable, the mixture fraction and its variance, the same as that used in the previous work [25].

In the present study, the average grid size corresponding to the use of 2 million cells is about 0.0226 mm, and the chosen time step is 0.0001 s. Because the performance of the SSTKWSAS turbulent model does not have an explicit mesh size and time step dependency, the performance of the model is comparable to that of LES simulation using a time step of $1 \times 10^{-7}s$ [26]. The SSTKWSAS model-based simulation is performed on the Solon cluster, City University of London. Statistical data is collected using the last 0.6 s of 1.6 s of total simulation time, considering that the flame residence time is 0.01 s. Readers might refer to [25] to gain more information about the simulation settings, such as the solver, mesh independency, cell quality, etc.

## 4. Results and Discussions

The instantaneous contour plot of a typical swirling strength [42] equaling 2222.24/s, and the time-averaged reaction progress at $y = 0$ m, a horizontal cutting plane, are presented in Figure 2. The SSTKWSAS model resolves more swirling eddies in the main flame region of the combustor compared to the two URANS models. The reason is well-recognized in the combustion science community: that the widely used URANS models have limited mechanisms to resolve the turbulent spectrum up to high wavenumbers. However, with a von Karman length scale introduced into SSTKW to form the SSTKWSAS model, the latter model well resolves the turbulent spectrum and generates a precessing vortex core (PVC) with a larger radius [43]. The occurrence of the PVC is a result of the vortex breakdown phenomenon [44] concerning the intensive swirling flows in the center recirculation zones (CRZs). The CRZs predicted by SSTKWSAS are hence more intensive and larger than those obtained from the URANS-based modeling methods. This leads to a faster or larger reaction progress for the SSTKWSAS model, as shown in Figure 2f. The fully combusted region (reaction progress equal to 1) is larger than the region obtained from the URANS simulation.

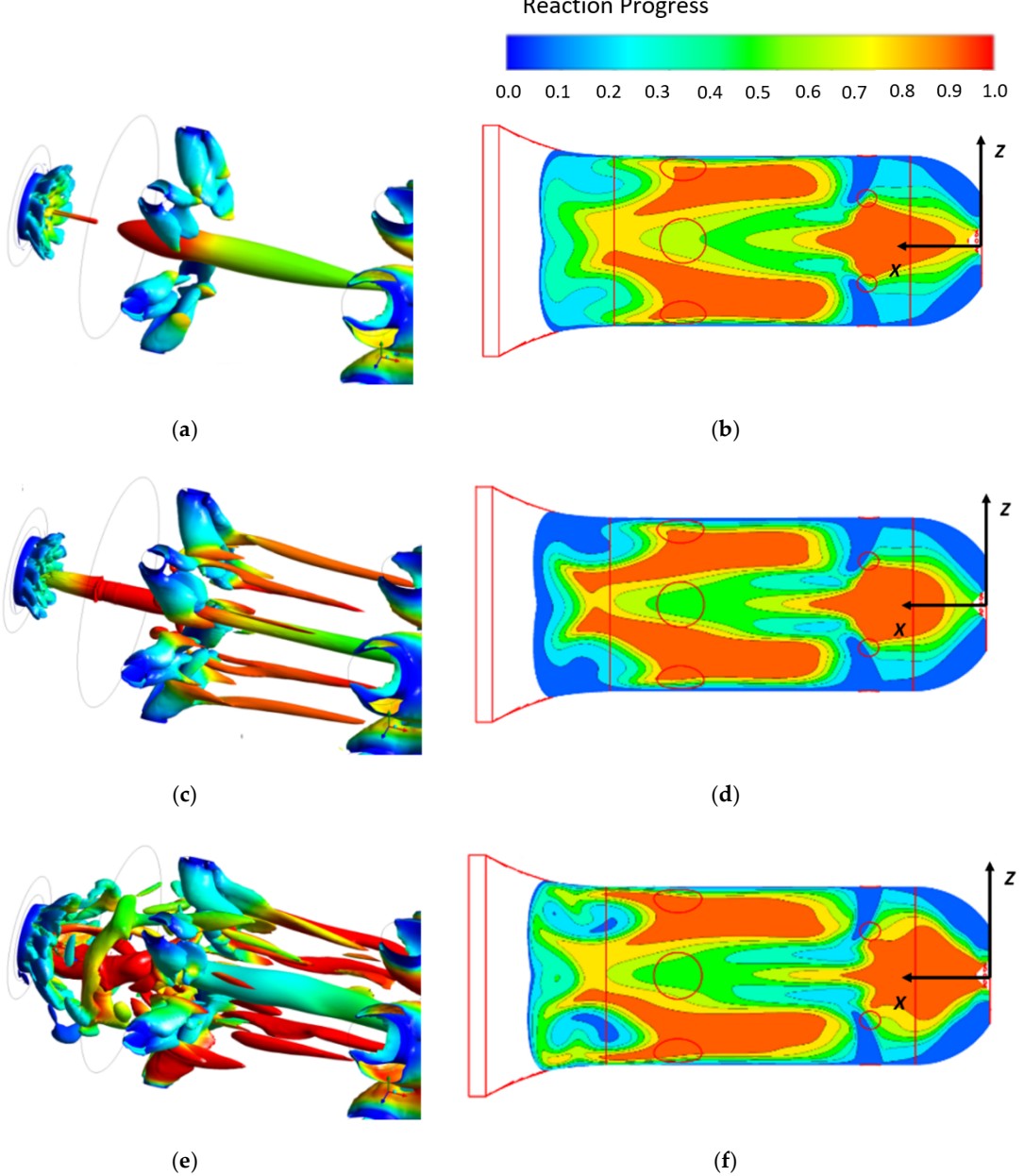

**Figure 2.** Instantaneous contour plots of a typical swirling strength equal to 2222.24/s (Left column) and the time-averaged reaction progress (right column). (**a**,**b**) SSTKW, (**c**,**d**) RSM. (**e**,**f**) SSTKWSAS.

Besides, Figure 3 presents the time-averaged mixture fraction distribution in the main flame region of the combustor. The fuel jets have clearly penetrated further towards the porous wall in the SSTKWSAS model, due to the CRZs inducing a higher momentum in its outer layers. The inner layers of the CRZs with negative axial velocity bring more oxygen from the primary holes back to the combustor head near the fuel holes, leading to a reduced mixture fraction distribution (the dark blue region). This also demonstrates that the CRZs obtained from the SSTKWSAS simulation are much stronger than those from the other URANS models-based simulations.

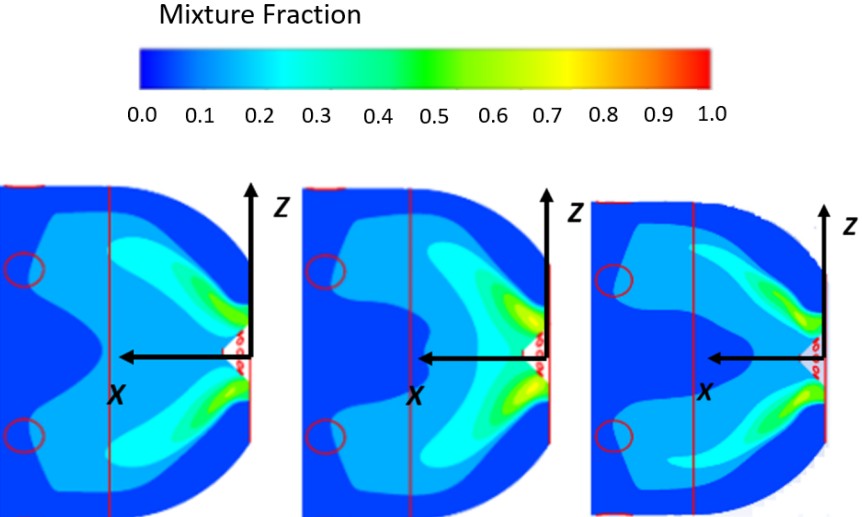

**Figure 3.** Time-averaged mixture fraction distribution in the main flame region of the combustor ($y = 0$ mm). Left: SSTKW, Middle: RSM, Right: SSTKWSAS.

The net effect of the intensive swirling eddies and the CRZs is a widening of the high-temperature zone in the main flame region of the combustor, as shown in Figure 4. The increased quantity of eddies resolved by the SSTKWSAS model act to prevent the high-temperature zone, predicted by the SSTKW model, from penetrating from the secondary region back into the primary region of the combustor. To the author's knowledge, such phenomena have not previously been highlighted because most researchers have preferred to focus only on simplified burners, with no sufficient important features of a practical gas turbine combustor, such as the primary and secondary holes. The penetrating flame predicted by the SSTK model does not only change the wall temperature distribution in the secondary region of the combustor, but also changes the hot fluid flow at the nozzle exit (the outlet of the combustor). This finding is believed to be important for turbine designers, though no detailed comparisons with experimental data are provided in the present study due to insufficient information about the geometric shape of the transition nozzle.

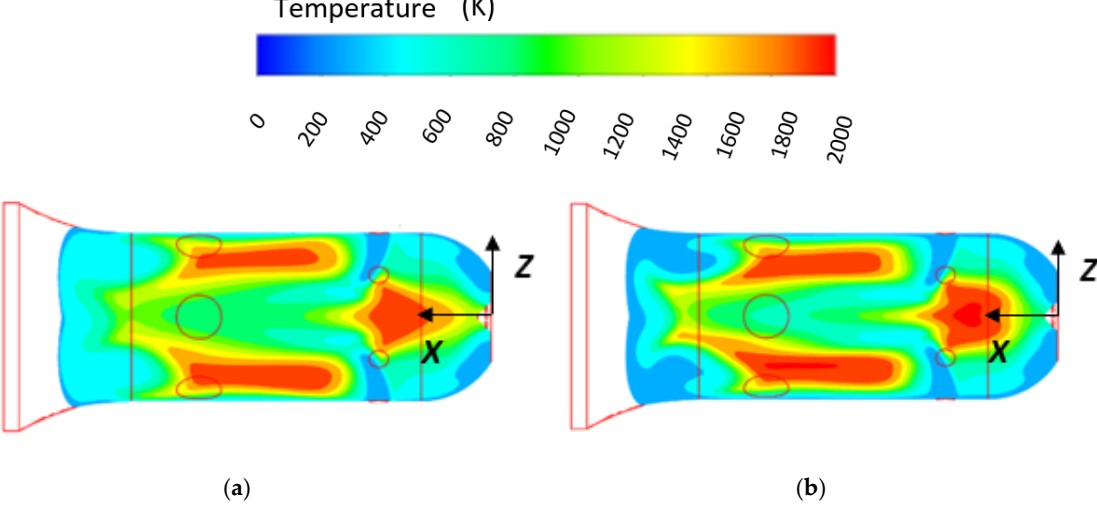

(**a**)          (**b**)

**Figure 4.** *Cont.*

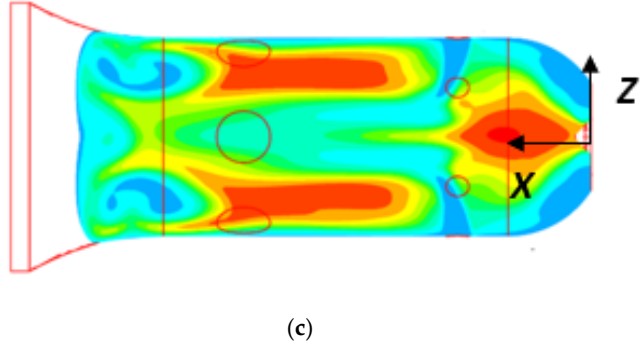

(**c**)

**Figure 4.** Time-averaged temperature contours on the horizontal midplane of the combustor ($y = 0$ mm): (**a**) SSTKW, (**b**) RSM, (**c**) SSTKWSAS.

In fact, Figure 5 shows that in the main flame region of the combustor, the SSTKWSAS model-based data fits reasonably well with the experimental data. The result supports the speculation made in our previous work [25] that a scale adaptive simulation such as SSTKWSAS may provide a more accurate representation of the flame behavior within the combustor head when the model is combined with a partially premixed finite-rate combustion model. An accurate representation of the flame behavior and its corresponding temperature in the main flame region is of the utmost importance in designing advanced gas turbines, and has been acknowledged to be difficult due to the insufficient validation of turbulent combustion models in a practical, industrial-used gas turbine combustor configuration. Moreover, in Figure 5, the less accurate LES result from Di Mare et al. [26] is due to the two reasons discussed in [25]: the use of an inappropriate, non-premixed, infinitely fast chemistry combustion model, and the lack of experimental data at the primary holes. In addition, an interesting phenomenon is observed in Figure 5b, such that the distribution of mixture fractions in the primary region is very similar when comparing the two predictions from SSTKWSAS and LES. This demonstrates that the SSTKWSAS model has an equal or at least similar ability to resolve the turbulent spectrum to that of the LES model, and can be considered a trustfworthy model to be employed in future industrial combustor simulation. It must be noted that, despite a similar mean mixture fraction distribution being observed, the two turbulent models do not present the same temperature distribution, clearly due to the coupling with different combustion models.

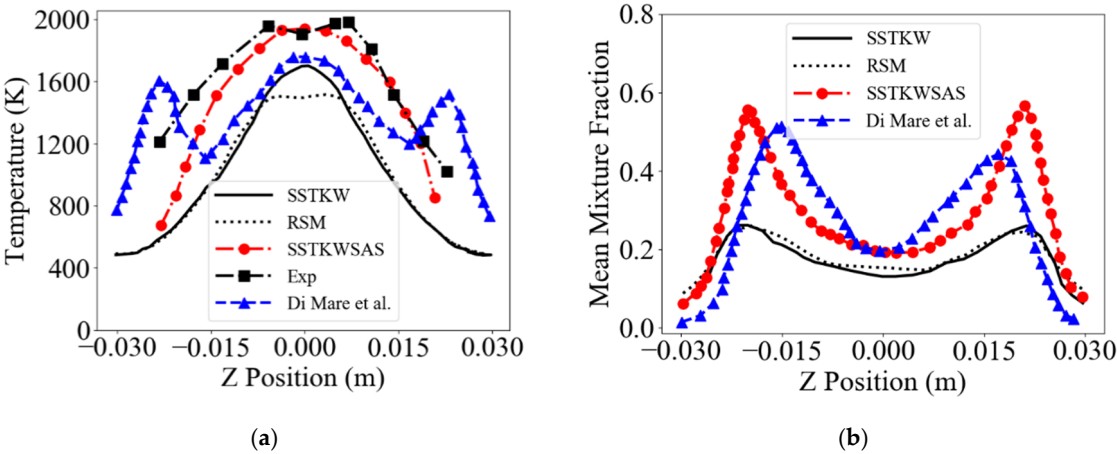

(**a**)　　　　　　　　　　　　　　　　　　　　　(**b**)

**Figure 5.** Profiles of time-averaged temperature and mixture fraction on the horizontal midplane of the combustor (at $x = 20$ mm). (**a**) Temperature Profiles, (**b**) Mixture fraction Profiles.

Figure 6 shows that in the main flame region of the combustor, the species mole fractions predicted by SSTKWSAS outperform those obtained from the URANS models and even the LES model-based

works [26]. The less accurate prediction of $CO_2$ level may imply that the chemistry closure method employed is not suitable for the slow CO oxidization reaction.

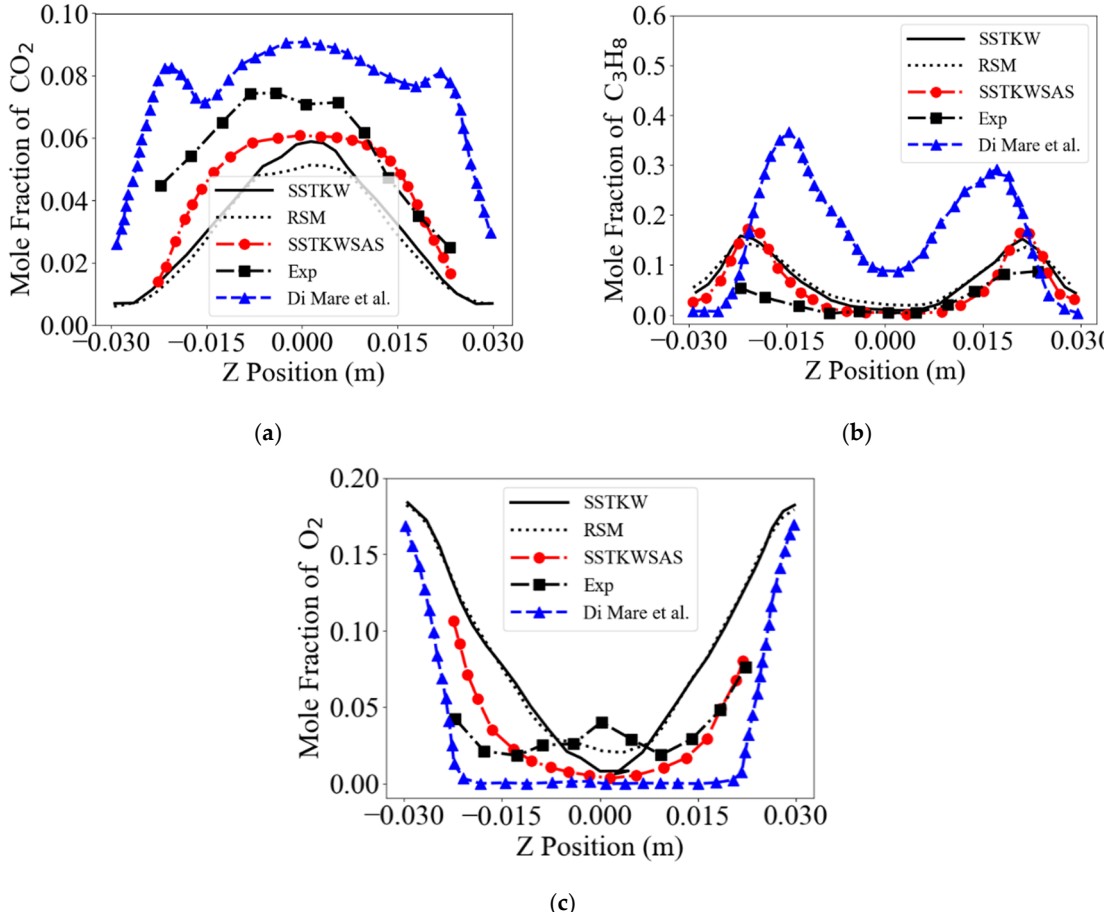

**Figure 6.** Profile of species mole fraction in the horizontal midplane of the combustor ($x = 20$ mm). (**a**) Mole fraction of $CO_2$, (**b**) Mole fraction of $C_3H_8$, (**c**) Mole fraction of $O_2$.

Finally, it is worth emphasizing that the chosen time step of 0.001 s for SSTKWSAS the modeling method is far larger than that used by Di Mare et al. in their LES work [26]. Hence, the present simulation requires far less computational power compared to the power requirement of the LES simulation. Due to the lack of information, such as software design, methodologies used etc., from the literature, more detailed comparisons of computational cost cannot be provided, but the reduction in the total time iterations used for the present study clearly represent a huge saving of computational power.

## 5. Conclusions

A scale-resolving simulation of a propane-fuelled industrial gas turbine combustor using finite-rate tabulated chemistry is performed, and the result is compared with that obtained from the two widely used URANs models, the SSTKW and RSM models. The chosen geometry includes important features of a commercial industrial-used gas turbine combustor, and provides an excellent test case to demonstrate the advantages of coupling a spectrum-resolving turbulent model, called SSTKWSAS, with a partially premixed, finite-rate chemistry-based combustion model in a very complex, multi-jet, 3D swirling flow environment. To reduce the computational time, a tabulated chemistry method is employed in conjunction with the SLFM method to simplify the employed comprehensive chemical kinetics. The turbulent strain-induced local chemical non-equilibrium is considered by taking into account the so-called scalar dissipation rate when the flamelets library is tabulated, and the pre-PDF

method is chosen as a bridge to cover the interaction between the turbulence and combustion. The main contributions and findings of the present study are:

(1) The spectrum-resolving SSTKWSAS model-based simulation is performed for the first time to predict the turbulent combustion in a practical industrial-used gas turbine combustor, by using a partially premixed, finite-rate chemistry combustion model.

(2) The SSTKWSAS model-based simulation outperforms the URANS-based simulation by having provided a more accurate temperature and species concentration in the main flame region of the combustor, contributing to its ability to resolve turbulence eddies at high wavenumbers.

(3) The inaccurate LES prediction of turbulent combustion in the main flame region of the combustor found in the literature is improved via the solutions discussed and provided.

Overall, the present study emphasized the dominant advantages of employing a more efficient and numerically accurate spectrum-resolving simulation model, SSTKWSAS, in simulating a 3D swirling flame developing in a complex, multi-jet flow environment. The SSTKWSAS model-based simulation was performed on the Solon cluster, City University of London. With 20 Computer Processing Units (CPUs) used, the total wall-clock time of 160 h was spent achieving one prediction on 2 million mesh cells. The total wall clock time of 26,432 h, using 64 CPUs, for one LES prediction of the same combustor available in the literature is unaffordable by most industries (1 million cells), though the prediction was done in 2004 [26].

**Author Contributions:** K.Z.: Methodology, Formal analysis, Writing—Original Draft, Writing—Review & Editing; A.G.: Conceptualization, Writing—Review & Editing, Supervision; J.M.N.: Writing—Review & Editing, Supervision, Funding acquisition. All authors have read and agreed to the published version of the manuscript.

**Funding:** This research received no external funding. However, partial financial support has been received from City, University of London.

**Acknowledgments:** The authors would like to thank the City, University of London for the HPC and financial supports for this research.

**Conflicts of Interest:** The authors declare that there is no conflict of interest.

## Nomenclatures

| | |
|---|---|
| BCD | Bounded central differencing |
| CRZ | Center recirculation zone |
| CV | Control volume ($m^3$) |
| C | Progress variable |
| $C_d$ | Discharge coefficient |
| d1, d2 | Inner and outer diameter of swirler ($m$) |
| b | Blockage factor |
| DES | Detached eddy simulation |
| $D_k$, $D_\omega$ | Dissipation of $k$ and $\omega$ |
| $E_\omega$ | Cross-diffusion term |
| EDC | Eddy dissipation concept |
| $G_k$, $G_\omega$ | Generation of $k$ and $\omega$ |
| $k$ | turbulent kinetic energy ($m^2 \cdot s^{-2}$) |
| $L_t$ | Turbulent length scale ($m$) |
| $L_{vk}$ | Von Karman length scale ($m$) |
| LES | Large eddy simulation |
| $\dot{M}_{air}$ | Mass flow rate of air ($Kg \cdot s^{-1}$) |
| $\dot{M}_{propane}$ | Mass flow rate of propane ($Kg \cdot s^{-1}$) |
| PDF | Probability density function |
| q | Reacting flow quantity |
| RSM | Reynolds stress transport model |

| $S$ | Swirl number |
|-----|--------------|
| SAS | Scale adaptive simulation |
| SLFM | Steady laminar flamelet modeling |
| $T_{inlet}$ | Inlet temperature of mixtures ($K$) |
| $U_t$ | Turbulent flame speed ($m \cdot s^{-1}$) |
| URANS | Unsteady Reynolds-averaged Navier–Stokes |
| ZTFSC | Zimont turbulent flame speed closure |

Greek symbols

| $\alpha_\theta, \alpha_c$ | Turbulent Prandtl and Schmidt number |
|---------------------------|--------------------------------------|
| $\omega$ | Vortex frequency ($s^{-1}$) |
| $\Delta$ | Mesh size ($m$) |
| $\rho, \rho_u$ | Fluid, and unburnt mixture density ($Kg \cdot m^{-3}$) |
| $\mu_{eff}$ | Effective viscosity ($kg \cdot m^{-1} \cdot s^{-1}$) |
| $\kappa$ | Von Karman constant |
| $\dot{\omega}_c$ | Chemical reaction rate ($Kg \cdot m^{-3} \cdot s^{-1}$) |
| $\eta$ | Turning efficiency |
| $\theta'$ | Geometric real angle (degree) |

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
