# Peer review of "Scale-Resolving Simulation of a Propane-Fuelled Industrial Gas Turbine Combustor Using Finite-Rate Tabulated Chemistry"

_fluids, doi:10.3390/fluids5030126_

Round 1

Reviewer 1 Report

  • Typo: Line 132.
  • Page 4: Can authors explain what definition of reaction progress variable is used? In a premixed combustion, we know what mixture composition at the unburnt stage and it is easy to define a reaction progress variable and track how reactants are transforming into products. Note that, here we know initial mixture. But, in a non-premixed combustion, mixture fraction is based on how fuel and oxidizer jet are diffusing into each other at a point and at that time instant. At the same mixture fraction, the diffusion of fuel and oxidizer are different at the pure mixing line ( we use non-reacting mixture fraction theory to compute the pure mixing line) and at other laminar flamelet solutions. So, the initial mixture is keep changing at every flamelet. Hence, every flamelet must have a C=0 and C=1. It seems, the authors are defining non reacting flamelet as C=0 and C=1 for fully burnt flamelet. Though, this approach is being used by other researchers as well, still the reviewer feels that it is not a correct approach.
  • An appropriate turbulent combustion model must be chosen based on the possible ratio between turbulence timescale to the chemical timescales. Does, the authors perform such analysis before choosing SLFM and ZTFSC models to simulate the presented combustor?
  • 7: "dot" is missing on source term.
  • Eq7: The source term is non-linear. Generally, it will have a maximum value close to C=0.8. This is where CO starts converting into CO2. I would like to request authors to substantiate why this model is chosen in this study.
  • Is the X-axis is coinciding the axis of the combustor? If so, represent it in the Fig. 3 accordingly. The results section has both -ve and +ve coordinates, so the axes representation might confuse the reader.
  • Do the authors have the experimental uncertainty values for Figure 5?

Reviewer 2 Report

The present paper compares and demonstrates the ability of the SSTKWSAS (shear stress transport, k-omega, scale adaptive simulation) model to better predict the turbulent combustion behavior than the SSTKW (shear stress transport, k-omega) and RSM (Reynolds stress transport) models. In addition, requiring significantly less computational resources than LES (large eddy simulation), the SSTKWSAS model is more accurate. This can be observed when comparing the numerical results with experimental data.

The following points should be however clarified/explained: 

What software was used to perform the simulations? Was the mesh independence study carried out? What kind of grid was used, structured, hexahedral? Why k-omega turbulence model was chosen instead of k-epsilon? What is the Reynolds number of the flow regime? What about the NOx emissions? How well the model can predict the NOx behavior? These questions should be answered and discussed/included in the paper.

Minor comments:

If you use abbreviations in the Abstract section, these should be defined, i.e., “shear stress transport, k-omega, scale adaptive simulation” (SSTKWSAS) (line 14) and “unsteady Reynolds-averaged Navier-Stokes” (URANS) (line 22).

Further, the following sentence should be revised (lines 12-16, too long, missing punctuation). Another example is (lines 34-37). Please improve.

All abbreviations should be mentioned in the text at their first use, e.g., SSTKWSAS (line 58), ITS (line 57), SSTKW (line 74), ZTFSC (line 118), CPU (line 305), or AFR (air-to-fuel ratio in Table 1).

Please complete (lines 103-106): all symbols in Equations (1-3), e.g., ρ, μ, Ew, etc., should be defined.

Please be specific, e.g., “with Equation (2) added to … Equation (1)” instead of “with the 2nd one added to remedy the defect of the 1st one” (lines 130-131).

Typing errors should be corrected before re-submission, e.g., “chtheemical” (line 132). Please proofread the manuscript.

Please be consistent, e.g., “g/s” (Table 1) vs. “kg/s” (Table 2).

Please correct: should be “10−7”! (line 184)

Some content is outside the margins, see lines 79-85 and 294-300.

Please include a list of all abbreviations/symbols at the end of the manuscript.
